# Molecular and Cytogenetic Identification of Wheat-*Thinopyrum intermedium* Double Substitution Line-Derived Progenies for Stripe Rust Resistance

**DOI:** 10.3390/plants12010028

**Published:** 2022-12-21

**Authors:** Guangrong Li, Qiheng Chen, Wenxi Jiang, Ahui Zhang, Ennian Yang, Zujun Yang

**Affiliations:** 1School of Life Science and Technology, University of Electronic Science and Technology of China, Chengdu 610054, China; 2Crop Research Institute, Sichuan Academy of Agricultural Sciences, Chengdu 610066, China

**Keywords:** chromosome translocation, ND-FISH, Oligo-FISH painting, rust resistance, *Thinopyrum intermedium*, wheat

## Abstract

*Thinopyrum intermedium* (2n = 6x = 42, JJJ^S^J^S^StSt) has been hybridized extensively with common wheat and proven to be a valuable germplasm source for improving disease resistance and yield potential of wheat. A novel disease-resistant wheat-*Th. intermedium* double substitution line X479, carrying 1St(1B) and 4St-4J^S^ (4B), was identified using multi-color non-denaturing fluorescence in situ hybridization (ND-FISH). With the aim of transferring *Thinopyrum*-specific chromatin to wheat, a total of 573 plants from F_2_ and F_3_ progenies of X479 crossed with wheat cultivar MY11 were developed and characterized using sequential ND-FISH with multiple probes. Fifteen types of wheat-*Thinopyrum* translocation chromosomes were preferentially transmitted in the progenies, and the homozygous wheat-1St, and wheat-4J^S^L translocation lines were identified using ND-FISH, Oligo-FISH painting and CENH3 immunostaining. The wheat-4J^S^L translocation lines exhibited high levels of resistance to stripe rust prevalent races in field screening. The gene for stripe rust resistance was found to be physically located on FL0–0.60 of the 4J^S^L, using deletion lines and specific DNA markers. The new wheat-*Th. intermedium* translocation lines can be exploited as useful germplasms for wheat improvement.

## 1. Introduction

Common wheat (*Triticum aestivum* L., 2n = 6*x* = 42, AABBDD) represents about 30% of the daily carbohydrate source and the primary source of protein for the world’s population [1]. The stripe rust (yellow rust), caused by *Puccinia striiformis* f. sp. *tritici* (Pst), is an important disease that occurs in most wheat-growing regions [2]. The areas affected by stripe rust epidemics have increased under both rain-fed and irrigated high-input conditions of many countries in recent years [3,4]. Considerable numbers of stripe rust resistance genes have been identified and used in wheat breeding [5,6]. However, due to limited numbers of effective resistance genes exploited in wheat cultivars and constantly evolving new pathogen races capable of overcoming deployed resistance genes, there is an urgent need to discover and exploit new sources of resistance [3]. Solutions to this disease problem affecting wheat production may be found in new genetic material derived from the wide hybridization of wheat with uncultivated species [7].

*Thinopyrum intermedium* consists of two subspecies, namely intermediate wheatgrass, ssp. *intermedium*, and a close pubescent relative, ssp. *trichophorum.* It is an important source of genetic variability for improving cultivated wheat, and carries novel and high levels of resistance to several wheat fungal diseases [8,9,10]. The novel genes for resistance to leaf, stem, yellow rust, and powdery mildew (*Lr38*, *Sr44*, *Yr50*, *Pm40*, and *Pm43*) were transferred to the wheat genome from *Th. intermedium* [11,12,13,14,15]. Conner et al. [16] reported that *Th. intermedium* ssp. *trichophorum* displayed high levels of resistance to wheat diseases, including rusts, root rot and WSMV. Yang et al. [8] reported that wheat-*Th. intermedium* ssp. *trichophorum* partial amphiploid TE-3 had resistance to several foliar diseases and unique seed storage proteins. Later, a series of wheat-*Th. intermedium* ssp. *trichophorum*-derived addition lines involving chromosomes 1St, 7St, 2J^S^, 3J, 4J^S^ and 5J^S^ were identified to display high resistance to stripe rust [17,18,19,20], and a novel *Th. intermedium* St-chromosome-specific HMW-GS gene was also transferred to wheat [21]. Therefore, *Th. intermedium* chromosomes possessing *Yr* genes should be further investigated for their potential use in breeding practice. 

Li et al. [22] isolated a double disomic substitution 1St(1B) + 4St–4J^S^ (4B) line, X479, and suggested that X479 is potentially useful for breeding for stripe rust resistance. In this study, the progenies from the cross between X479 and MY11 were examined to determine the genetic recombination and translocation between wheat and *Th. intermedium* chromosomes using non-denaturing fluorescence in situ hybridization (ND-FISH), Oligo-FISH painting and molecular markers. Genetic analysis for the physical location of *Th. intermedium* chromosome region carrying stripe rust resistance was also included.

## 2. Results

### 2.1. Sequential ND-FISH Patterns of Thinopyrum Chromosomes in X479

Sequential ND-FISH with probes Oligo-B11, Oligo-pDb12H, Oligo-K288, Oligo-pSc119.2, Oligo-pTa535, Oligo-pTa71, Oligo-pSt122, and a new probe, Oligo-744, were used to detect the chromosome constitution of X479 (Figure 1). Probes Oligo-B11 and Oligo-pDb12H allowed us to identify *Th. intermedium* and J^S^ genome [23,24], while Oligo-k288 and Oligo-D detect wheat AB-genome and D-genome chromosomes [23,25]. The sequential ND-FISH with those probes revealed that X479 carried 24 AB-genome chromosomes, 14 D-genome chromosomes, and four *Th. intermedium* chromosomes, including 1St and 4St-J^S^ (Figure 1b). The probes Oligo-pSc119.2 and Oligo-pTa535 confirmed that wheat chromosome 1B and 4B pairs were substituted by a pair of *Th. intermedium* 1St and 4J^S^-St chromosomes (Figure 1c), which is consistent with the previous GISH study by Li et al. [22]. The chromosome 1St had the signals of Oligo-B11 across the entire chromosome, and Oligo-pTa71 had one strong site in the terminal region and one site close to the centromeric region of 1StS, while the Oligo-pSt122 only hybridized to terminal regions of 1StL (Figure 1d,e). Chromosome 4St-J^S^ displayed distinct hybridization of Oligo-B11 and Oligo-pDb12H, showing the translocation on the short arm (Figure 1a). The Oligo-744 had strong signals at the telomeric region of the short arm, and two sites on the distal and proximal regions of the long arm of chromosome 4St-J^S^ (Figure 1d). Meanwhile, ND-FISH hybridization of Oligo-744 was tested for the chromosome 4J^S^ and 4St in wheat-*Th. intermedium* partial amphiploid [26], and we found that the chromosome 4St had strong signals at the telomeric region of the short arm, while the 4J^S^ had two sites on the centromeric and telomeric regions of the long arm (Figure 1f,g). The comparative ND-FISH patterns confirmed that the Oligo-pSc119.2 sites of 4J^S^S had been replaced by the Oligo-744 sites of 4StS to form the chromosome 4St-J^S^ in X479 (Figure 1g). Based on the ND-FISH hybridization patterns of multiple probes on the chromosomes of X479 and other wheat genotypes, we will be able to trace the transmission and variations of chromosomes 1St and 4St-J^S^ in different wheat background.

### 2.2. Chromosome Transmission in Progenies between X479 and MY11

In order to transfer novel resistance from X479 to wheat, the offspring of the hybrid between X479 and MY11 were traced with sequential ND-FISH using multiple probes, including Oligo-pSc119.2, Oligo-pTa535, Oligo-pTa71, Oligo-pSt122, Oligo-B11, Oligo-pDb12H, and Oligo-(GAA)_7_. A total of 248 plants of the F_2_ generation were studied using ND-FISH, and the representative plants with 1St, 4St-J^S^ and the wheat-1St, wheat-4St-J^S^ translocation chromosomes were indicated in Figure 2. We found that total of 78 of these plants (31%) contained chromosomes with structural changes, of which 26 plants contained a telosomic chromosome, while 51 had chromosome translocations, and one had telosomics and translocations. A total of 325 F_3_ plants were also investigated for chromosomal rearrangements. As shown in Appendix A, a total of 20% of plants contained the 63 types of chromosome translocations, and 25 different types of either broken or translocated chromosomes were also observed. Among them, translocations occurred more frequently between wheat and 4St-J^S^ (36.51%) than those between wheat and 1St (25.40%). It is interesting to find that the translocations of 4St-J^S^.4BL, iso-chromosome 4St-J^S^.4St-J^S^, 1StS.1DL and 1DS.4J^S^L were observed at high rates. The variation of chromosomal constitution in the 1St- and 4St-J^S^-derived progenies indicates that wheat chromosomes 4B and 1B may be preferentially involved in the translocations. The sequential ND-FISH using multiple probes was useful for precise localization of the translocations between wheat and *Th. intermedium* chromosomes.

### 2.3. Characterization of the Homozygous Wheat-Th. intermedium Translocations

We set out to recover plants homozygous for the wheat-1St and wheat-4J^S^ translocations from the F_4_ generation of hybrids between X479 and wheat. Homozygous wheat-1St transfers including 1BS.1StL and 1BL.1StS were also obtained in 11–25% of plants. Lines M913 and M971 were identified with a pair of chromosomes 1BS.1StL (Figure 3a,b), 1StS.1BL (Figure 3c,d) through sequential ND-FISH, respectively. The line M913 had a homozygous 1BS.1StL chromosome with the transmission rate of 16.7%, while M971 had the homozygous 1BL.1StS translocation with 100% transmission in its selfed progenies. This indicates that the 1BL.1StS translocation may have better complementation than that of 1BS.1StL in a wheat background. The wheat-4J^S^ translocation line M942 having 1BS.4J^S^L and line M965 having 1DS.4J^S^L chromosomes were observed with sequential ND-FISH and Oligo-FISH painting (Figure 4a–d). The Oligo-FISH painting, using the probe Synt1, generated distinct red signals on the homoeologous group 1 chromosomes of wheat 1A, 1B, 1D and *Th. intermedium* 1St, and Synt4 hybridizes to the homoeologous group 4 chromosomes 4A, 4B, 4D and 4St-J^S^. The translocation chromosomes 1BS.4J^S^L (Figure 4b) and 1DS.4J^S^L (Figure 4d) were indicated with the two colors corresponding to the hybridization signals of Synt1 and Synt4. These wheat-1St translocation chromosomes exhibited higher transmission rates than those with 4J^S^L, involving non-homologous chromosomes in the latter generations of selfed progenies.

ND-FISH with the wheat centromere-specific repeat Oligo-CCS1 probe (Figure 5a) revealed that line X479 contained two pairs of chromosomes devoid of hybridization signals. Sequential ND-FISH using Oligo-pTa71 and Oligo-pSt122 confirmed that these two pairs of chromosomes are chromosomes 1St and 4St-J^S^ (Figure 5b), indicating that the centromeric repeats varied between the species of wheat and *Th. intermedium*. When Immuno-FISH analysis was conducted using antibodies against wheat CENH3 [27], all the chromosomes of X479 showed the CENH3 signals in the centromeric regions, implying that CENH3 binds to centromeric positions of both the wheat and *Thinopyrum* chromosomes (Figure 5c). The sequential ND-FISH confirmed identification of the *Thinopyrum* chromosomes 1St and 4St-J^S^ (Figure 5d). To test for the amount of CENH3 accumulated at the centromeres of the chromosomes 1StS.1BL, 1BS.1StL and 4BS.4J^S^L, we used immunostaining with the anti-CENH3 antibody and sequential ND-FISH to the metaphase chromosomes of translocation lines (Figure 5e). The analysis of CENH3 immunostaining signals in 1StS.1BL, 1BS.1StL and 4BS.4J^S^L was conducted to estimate the amounts relative to that of averaged wheat chromosomes in the same metaphase cells. We found that the overall CENH3 accumulation on 1StS.1BL was higher than that of 4BS.4J^S^L and 1BS.1StL chromosomes. The results demonstrate that the wheat-1St and 4J^S^L translocations may accumulate enough content of CENH3 protein in the centromere regions during mitosis for reliable chromosome transmission.

### 2.4. Validation of Wheat-Th. intermedium Chromosome Deletion using ND-FISH and Molecular Markers

The Oligo-pDb12H sequences were used for BLAST searching of the *Thinopyrum intermedium* reference genome sequences v2.1 by B2DSC web sites [28]. The J^S^ chromosomes in the reference genome were determined by the distribution of Oligo-pDb12H (Appendix A), and consequently predicted to have 2074–4868 copies of Oligo-pDb12H, and chromosome 4J^S^ (the chromosome 12 or 4J^VS^ of Cseh et al. [29]) had 2592 copies. The result is consistent with the ND-FISH study, which indicated the level of the enrichment of Oligo-pDb12H sites in the J^S^-chromosomes. A total of 55 4VL-specific CINAU markers (Appendix A) [30] were physically localized on 176–435Mb of *Thinopyrum intermedium* V2.1, as shown in Figure 6. A line M931 was identified to have a pair of 4St-4J^S^ deletion chromosomes, and ND-FISH results indicated that M931 had lost the Oligo-B11 signals in the long arm of the 4J^S^L, with the break point estimated about FL0.6 of 4J^S^L (Appendix A). The DNA of the lines CS, MY11, X479 and M931 was used to assign the markers onto the deletion chromosomes in line M931. As indicated in Figure 6c, the CINAU1337 was located at 308.86 Mb, and proximal markers on the centromeric region were present in M931, but CINAU1266 located at 311.86 Mb distal to the telomeric regions was amplified only in X479. Therefore, the estimated physical position of FL0.6 of 4J^S^L in M931 may be localized between the 308.86 to 311.86 Mb genomic region of 4J^S^ (Figure 6). 

### 2.5. Stripe Rust Resistance Observations of Wheat-Th. intermedium Introgression Lines 

In order to evaluate the contribution of the two wheat-*Th. intermedium* translocated chromosomes to agronomic characteristics in the progeny of X479 and MY11, the resistance responses of each plant to stripe rust were observed among the plants karyotypically identified with ND-FISH. Infection type responses of X479 and its derived progenies were measured at the adult plant stages in the field. As shown in Figure 7, X479 was highly resistant to stripe rust (IT = 0), while MY11 was highly susceptible (IT = 4). Among its derived progenies, plants with the different translocation chromosomes 1BS.4J^S^L (M942) and 1DS.4J^S^L (M965), and the 4St-J^S^ deletion (M931) showed high resistance (including some plants with hypersensitive reactions after rust inoculation, IT = 0 to 0;). The plant M102 containing only the 4J^S^S translocation (4St-4J^S^S.5AL) was susceptible to the rust races (Figure 7). These results indicate that the stripe rust resistance was from the *Th. intermedium* and may be physically localized on FL0–0.60 of 4J^S^L.

## 3. Discussion

The techniques of fluorescence in situ hybridization (FISH) and genomic in situ hybridization (GISH) have shown significant impacts on the cytogenetic studies in bread wheat and related species [31,32]. The alien chromosome segments, when introgressed into wheat, can be successfully detected using GISH and FISH [12,33,34]. Based on the GISH patterns using *Ps. strigosa* genomic DNA (St genome) as a probe, Li et al. [22] demonstrated that two pairs of *Thinopyrum*-derived chromosomes (St genome and St–J^S^ translocated chromosomes) substituted for two pairs of wheat chromosomes in X479. The FISH, with the J^S^ chromosome-specific repetitive sequence pDb12H, revealed the translocation of St–J^S^ chromosomes in X479. In the present study, ND-FISH using Oligo-B11 with Oligo-pDb12H allowed us to identify the constitution and the translocations of *Th. intermedium* chromosomes in X479. Chromosome 4St-J^S^ displayed distinct ND-FISH hybridization patterns of the probes Oligo-744 and Oligo-pSc119.2, showing that the formation of the chromosome 4St-J^S^ in X479 involved the Oligo-pSc119.2 sites of 4J^S^S replaced by the Oligo-744 sites of 4StS at the telomeric region of the short arm. Therefore, our study demonstrated that ND-FISH techniques are highly effective for precise identification and mapping of wheat-alien chromosomal rearrangements, replacing the traditional FISH and GISH methods [24,25,26].

Stable transmission of alien chromosomes in wheat background is a fundamental requirement for utilizing genes located on these chromosomes in wheat breeding. Many studies have shown that alien chromosomes in wheat generally exhibit transmission behavior different from the background of the native wheat chromosomes [35,36,37]. For example, Whelan et al. [38] found that the *Agropyron elongatum* chromosome 6AgS in a wheat background was transmitted at a higher rate through the female gametes (40–50%) than through the male gametes (3%). Similarly, the recombinant chromosomes in different non-homoeologous wheat–*Dasypyrum villosum* 6V translocation lines could be transmitted through both the male and female gametes [39]. Recently, Wang et al. [40] studied the double substitution lines with chromosomes of barley 2H and *Thinopyrum intermedium* 2Ai#2 in different reciprocal cross combinations in wheat. Chromosome 2H showed a higher transmission rate than alien chromosome 2Ai#2 in the progenies. In the present study, we observed that the hybrids between the disomic substitution line of X479 and normal wheat resulted in the production of double monosomics for chromosomes 4B, 1B, 4St-J^S^ and 1St. Mis-division of chromatids in the hybrids generated mostly monosomic chromosomes in the subsequent progenies, which provided opportunities for the formation of translocations involving the linkage group 1 and group 4 chromosomes of wheat and *Thinopyrum*. Translocations occurring between wheat chromosomes and 4St-J^S^ were more frequent than those between wheat and 1St, indicating that the complementarity of 4St-J^S^ to 4B is lower than those of 1St to 1B. The broken and re-fused wheat-*Thinopyrum* chromosomes give rise to the homozygous complementary Robertsonian translocation of wheat-4J^S^, and 1St for gene localization of *Thinopyrum* chromatin in wheat background. 

The CENH3 is an essential component of the kinetochore complex of active centromeres, and the mitotic interphase cells are ideal for demonstrating co-localization of CENH3 and centromere repetitive DNA sequences, although the centromere repeats may change extremely rapidly [41,42]. In wheat-alien interspecific hybridization, the wheat-*Thinopyrum* partial amphiploids containing typical centromeric repeats were well co-localized with CENH3 [43]. In the present study, the substitution line X479 displayed chromosomal distribution of CENH3 in wheat and the CENH3 peptide generated distinct signals in the centromeric regions of 1St and 4St-J^S^ chromosomes (Figure 5a,c), as well as the progenies with 1StS.1BL, 1BS.1StL and 4BS.4J^S^L, which were detected using immunostaining experiments (Figure 5e). The observed CENH3 signals were co-localized with Oligo-CCS1 signals in wheat with roughly identical densities, although the Oligo-CCS1 repeats in the wheat A, B and D genome showed with divergent copies [28]. It is likely that centromere-associated DNA sequences are not fixed but are dynamic in their amounts and positions on chromosomes in Triticeae. Further studies will be performed to reveal the *Thinopyrum*-derived centromere incorporated CENH3 of wheat in the wheat-1St or wheat-4J^S^L hybrid centromere, as well as the mechanism of wheat-*Thinopyrum* translocated chromosomes for maintaining their chromosome stability with respect to evolutionary and breeding purposes [44,45].

Stripe rust is widely regarded as one of the most destructive wheat worldwide diseases that affects wheat production globally. Up to now, more than 84 designated wheat stripe rust resistance genes have been formally designated [6,46]. Among them, only *Yr50* has been transferred from *Th. intermedium* into cultivated wheat [15]. However, the stripe rust resistant genetic resources originating from *Th. intermedium* have been reported. Li et al. [19] identified a 2J^S^ addition line, two substitution lines of 4J^S^(4B) and 4J(4B), and a small 4BS.4BL-4JL translocation line using FISH and molecular markers. Their studies indicated that chromosomes 4J^S^ and 2J^S^ enhanced the resistance to stripe rust in the adult plant stage. Li et al. [20] developed a new wheat-*Th. intermedium* 3J disomic addition line (A1082) that is highly resistant to stripe rust, and a 4BS.5J^S^L translocation line (A5–5) that was moderately resistant to the stripe rust pathogens. In the present study, we identified the homozygous Robertsonian translocation 1BS.4J^S^L and 1DS.4J^S^L (Figure 7). The lines have been subjected to ^60^Coγ irradiation and *ph1b* mutation to make better use of the current materials and to obtain small-segment introgression lines with stripe rust resistance genes. The small segment translocation lines will be acquired and used in the wheat breeding programs as means of developing such lines. In addition, the subsequent target-sequence enrichment and sequencing (TEnSeq) pipeline [47] for EMS mutated wheat-4J^S^L lines are ongoing, with the aim of dissecting the novel resistance gene with the help of a further-updated genome project of *Th. intermedium* [29,48]. 

## 4. Materials and Methods

### 4.1. Plant Materials

The wheat-*Th. intermedium* ssp. *trichophorum* partial amphiploid TE1508 [18], a double 1St#2 (1B) plus 4St/4J^S^ (4B) substitution line X479 [22], as well as wheat cultivars Chinese spring (CS) and Mianyang11 (MY11) are maintained in our laboratory at University of Electronic Science and Technology of China. *Th. intermedium* PI440043 (StJ^S^J genomes, 2n = 6x = 42) were obtained from the National Small Grains Collection at Aberdeen, ID, USA.

### 4.2. Fluorescence In Situ Hybridization (FISH)

Seedling root tips at 2–3 cm were collected, and treated with nitrous oxide at 1.0MPa for two hours. They were fixed in 90% acetic acid for 10min. The washed root tips were digested with 1% pectolyase Y23 (Yakult Pharmaceutical, Tokyo, Japan) and 2% cellulase Onozuka R-10 (Yakult Pharmaceutical Tokyo, Japan) solution at 37 ℃ for 50 min. The suspension of root meristematic region cells was squashed, following the procedure of Tang et al. [49]. The protocol of non-denaturing FISH (ND-FISH) with synthesized oligo-nucleotide probes Oligo-pSc119.2, Oligo-pTa535, Oligo-pTa71, Oligo-pSt122, Oligo-B11, Oligo-pDb12H and Oligo-744 were used for identifying the wheat chromosomes and *Th. intermedium* chromosomes (Appendix A). The physical location and copy number of Oligos were predicted as described by Lang et al. [28]. The synthetic oligonucleotides were 5′ end-labeled with 6-carboxyfluorescein (6-FAM) for green or 6-carboxytetramethylrhodamine (Tamra) for red signals. Photomicrographs of FISH chromosomes were captured with an Olympus BX-53 microscope equipped with a DP-70 CCD camera.

After the ND-FISH, the chromosome squashes were prepared for sequential FISH by being washed twice with each 5 min by 0.1% Tween 20 in 2xSSC to remove the hybridization signals. The Oligo-FISH painting probed with the 45bp single-copy oligo pools Synt1 to Synt7 for wheat-barley linkage group specific was used to reveal the linkage group assignment for wheat and Triticeae chromatin [50]. The procedure of Oligo-FISH painting was followed the description by Li and Yang [51].

### 4.3. Molecular Marker Analysis

DNA was extracted from young leaves of Chinese Spring (CS), Mianyang 11 (MY11), X479 and their derived progenies using the SDS protocol [8]. Linkage group 4 of CINAU markers (Appendix A) [30] were searched for chromosomal physical locations using the database of *Thinopyrun intermedium* genome V.2.1 (https://phytozome-next.jgi.doe.gov/info/Tintermedium_v2_1, accessed on 21 January 2021). Polymerase chain reaction (PCR) was performed according to the description of Li et al. [19] in an Icycler Thermal Cycler (Bio-Rad Laboratories, Emeryville, CA, USA). The PCR products were electrophoresed on a 1.0% agarose gel, or 8% PAGE gel, as described by Hu et al. [17] and Yu et al. [52], respectively.

### 4.4. Sequential Immunolocalization-FISH

The anti-CENH3 antibody was an affinity-purified rabbit polyclonal antibody against two peptides as reported by Yuan et al. [27], and was generated by GL Biochem Ltd. (Shanghai, China). The immuno-localization for mitotic metaphase cells of antibodies against CENH3 (1:200) and the detection using the goat anti-rabbit Texas red (1:500; Sigma-Aldrich, St Louis, MO, USA) were performed, as described by Han et al. [53]. The images were collected with the BX53 Motorized System Microscope (Olympus). The slides were treated on fixation solution (3 ethanol:1 acetic acid) before ND-FISH protocol was performed for chromosome identification as above-mentioned.

### 4.5. Rust Resistance Tests

To evaluate the rust resistance of X479 and its derived progenies, we used a mixture of stripe rust (*P. striiformis* f. sp. *tritici*, Pst) races CYR32, CYR33 and CYR34. All materials were tested in the field at the Xindu Experimental Station, Chengdu, China during 2018–2022. Infection types (IT) were scored 14–15 days after inoculation when rust was fully developed on the susceptible check. The IT was recorded based on the 0–4 scale according to Bariana and McIntosh [54].

## 5. Conclusions

The precise identification of the double disomic substitution line X479 revealed the introgression of *Thinopyrum intermedium* 1St and 4St–J^S^ translocation using sequential ND-FISH. The progenies from the cross between X479 and MY11 were found to have extensive wheat-*Thinopyrum* chromosome recombination and translocation which were revealed using ND-FISH, Oligo-FISH painting and molecular markers. The chromosome translocations and deletions of the wheat-*Thinopyrum* introgression lines enabled the physical location of stripe rust resistance on the *Th. intermedium* chromosome 4J^S^L. These novel wheat-*Thinopyrum* introgression lines are great potential resources for wheat breeding of resistance.

## Figures and Tables

**Figure 1 plants-12-00028-f001:**
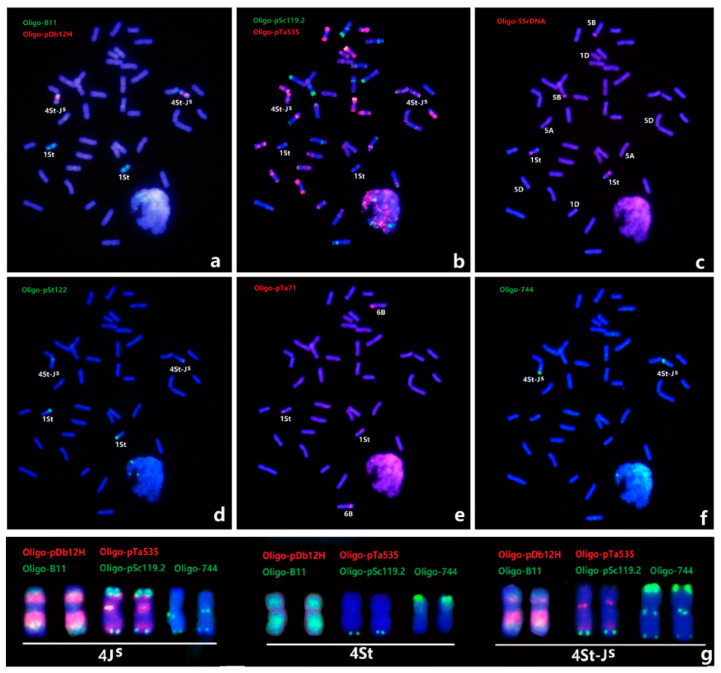
Sequential ND-FISH patterns of wheat-*Th. intermedium* double substitution line, X479. The probes for FISH are: (**a**) Oligo-B11 (green) and Oligo-pDb12H (red); (**b**) Oligo-pSc119.2 (green) and Oligo-pTa535 (red), (**c**) Oligo-5SrDNA; (**d**) Oligo-pSt122; (**e**) Oligo-pTa71; and (**f**) Oligo-744 (green). The karyotype of the 4J^S^, 4St and 4St-J^S^ chromosomes are shown using the probe combinations Oligo-B11 with Oligo-pDb12H (left), Oligo-pSc119.2 and Oligo-pTa535 (middle), and Oligo-744 (right), respectively (**g**).

**Figure 2 plants-12-00028-f002:**
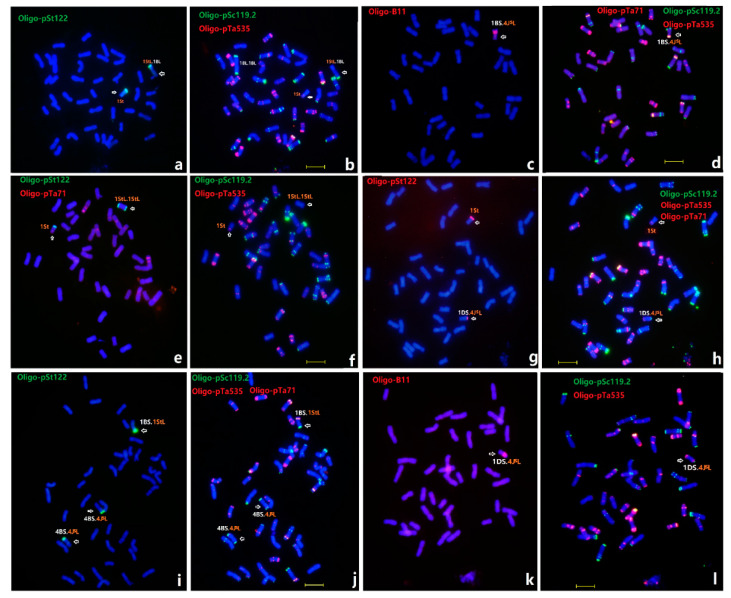
Sequential ND-FISH of F_2_ progenies of hybrids between X479 and MY11. The lines C423 (**a**,**b**), C32 (**c**,**d**), C451 (**e**,**f**), C429 (**g**,**h**), C436 (**i**,**j**), C460 (**k**,**l**) were indicated. The arrows indicated the introgressed *Thinopyrum* or wheat-*Thinopyrum* translocation chromosomes. Bars, 10 μm.

**Figure 3 plants-12-00028-f003:**
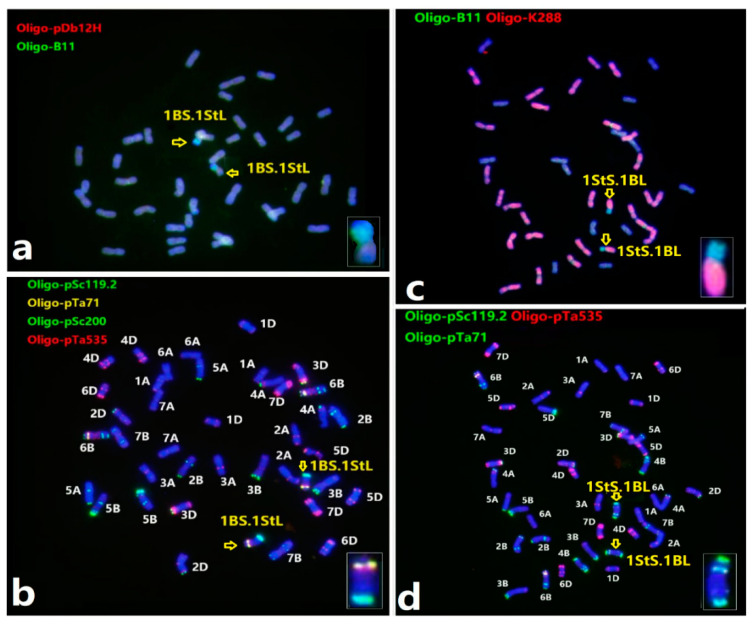
Sequential ND-FISH patterns showing wheat-*Th. intermedium* homozygous 1St translocation lines. (**a**,**b**) 1BS.1StL and (**c**,**d**) 1BL.1StS. The probes for FISH were: (**a**) Oligo-B11 (green) with Oligo-pDb12H (red); (**b**) Oligo-pSc119.2 (green) with Oligo-pTa535 (red); (**c**) Oligo-k288 (red) with Oligo-D (green); and (**d**) Oligo-pSc119.2 (green) with Oligo-pTa535 (red). The arrows indicated the translocation chromosomes.

**Figure 4 plants-12-00028-f004:**
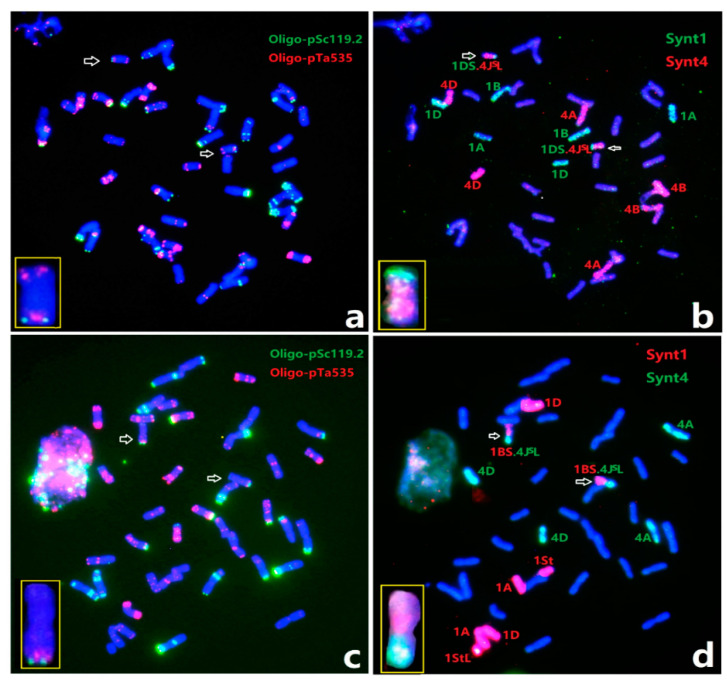
Sequential ND-FISH and Oligo-FISH painting of wheat-4J^S^L translocation lines, (**a**,**b**) 1DS.4J^S^L and (**c**,**d**) 1BS.4J^S^L, using the ND-FISH probes (**a**,**c**) Oligo-pSc119.2 with Oligo-pTa535 and Oligo-FISH painting probes (**b**,**d**) Synt1 and Synt4, respectively. The chromosomes indicated with arrows are extracted and put in the boxes to indicate the translocated chromosomes.

**Figure 5 plants-12-00028-f005:**
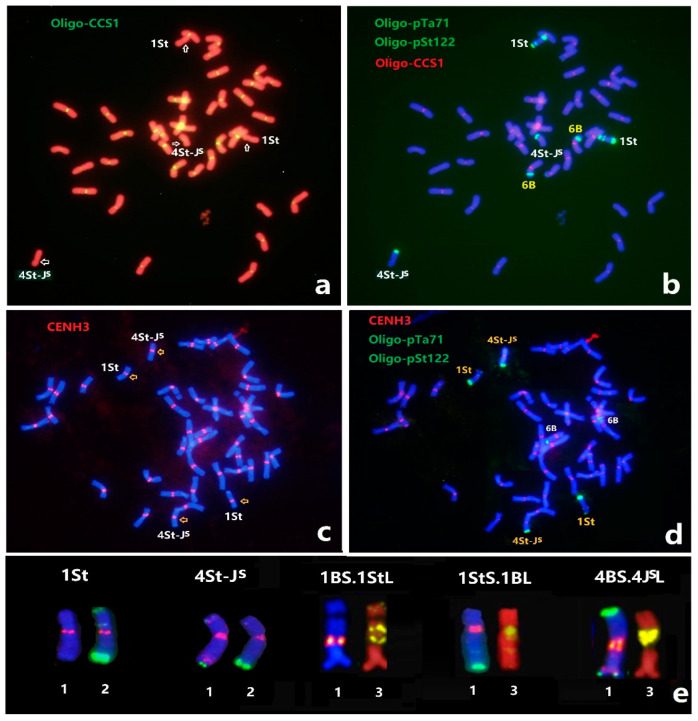
The centromeric repeats Oligo-CCS1 and anti-CENH3 location in X479 and derived translocations from immunostaining with the anti-CENH3 antibody and sequential ND-FISH. Sequential FISH of X479 by (**a**) Oligo-CCS1 and (**b**) Oligo-pTa71, Oligo-St122 and Oligo-CCS1. The arrows indicate the absent signals of Oligo-CCS1 in 1St and 4St-J^S^ chromosomes. (**c**) Sequential immunostaining of anti-CENH3 and (**d**) ND-FISH of X479 by Oligo-pTa71 and Oligo-pSt122. The arrows indicate the CENH3 signals presented in 1St and 4St-J^S^ chromosomes. (**e**) The FISH and anti-CENH3 signals for wheat-1St and wheat-4J^S^L translocation. The numbers 1–3 represent the combinations of Oligo-pSc119.2 with Oligo-pTa535 and CENH3, Oligo-pTa71 with Oligo-pSt122, and Oligo-(GAA)_7_, respectively.

**Figure 6 plants-12-00028-f006:**
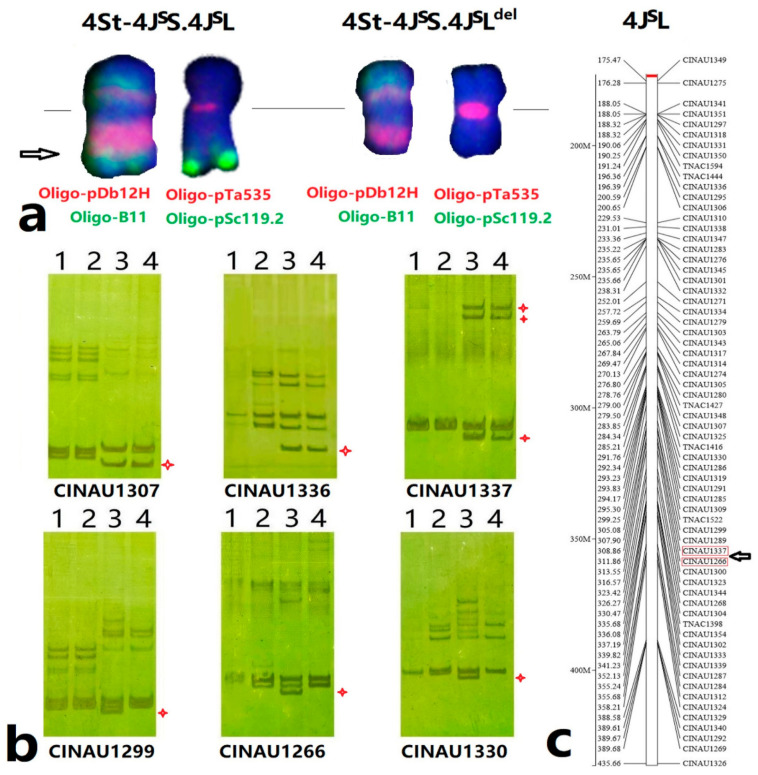
Physical location of the break point of 4St-4J^S^S.4J^S^L deletion detected using FISH and molecular markers. (**a**) The comparative FISH patterns of the 4St-4J^S^S.4J^S^L and 4St-4J^S^S.4J^S^L^del^. Arrow shows the break point location. (**b**) The PCR amplification of CINAU markers for the lines (1) CS, (2) MY11, (3) X479 and (4) M931. The stars show the 4J^S^L amplifications. (**c**) The 4VL-specific CINAU markers were blasted to locate on the 4J^S^L of *Thinopyrum intermedium* genome sequence of v2.1. The break point in 4J^S^L is shown by an arrow.

**Figure 7 plants-12-00028-f007:**
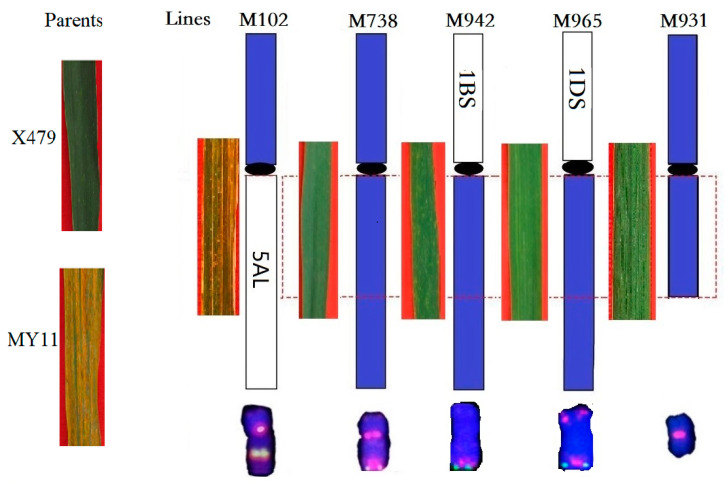
Physical location of wheat-*Th. intermedium* lines using rust resistance survey and ND-FISH. The left diagrams show the stripe resistance of leaves of parents and derived lines, and the chromosomes variation on the bottom with ND-FISH using probes Oligo-pSc119.2 (green) and Oligo-pTa535 (red).

## Data Availability

Data are contained within the article.

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
