# Peer review of "Molecular and Cytogenetic Identification of Wheat-Thinopyrum intermedium Double Substitution Line-Derived Progenies for Stripe Rust Resistance"

_plants, 2022, doi:10.3390/plants12010028_

Round 1

Reviewer 1 Report

Exceptionally nice work. Material section lacks some details and needs to be expended. 

Author Response

Exceptionally nice work. Material section lacks some details and needs to be expended.

Response: Thanks for the comment and suggestion. We agree to edit the relevant description and added detail primer sequences in Table S2.

Reviewer 2 Report

The present paper describes cytological identification of the progenies of wheat-Thinopyrum intermedium chromosome double substation line and localization of the gene for stripe rust resistance on the translocated chromosome. The reviewer considers that the quality of the experiment is valuable to be published in Plants. However, many mistakes and wrong usages of English words are found in the manuscript. 

1)   The reviewer pointed out the mistakes and gives corrections recommended as below, and would like to ask the authors to correct the manuscript and submit English review to check the expressions.

2)   Explanations for Figures should be checked precisely. It is important to indicate the translocated chromosomes. Especially, explanation for Figure 2 is not exactly mentioned. There are six pairs of pictures (a-l). However, there are no explanations on that. In the text, it is written that they are the pictures of Fplants, but in the explanation of Fig. 3, they are F3 plants. Which is correct?

Line 212: The symptoms of X479 and MY11 are not shown in Figure 7 as mentioned in the text.

3)   On the markers closely linked to the break point of the translocation, the sequences of the markers used in Fig. 6 for PCR amplification should be shown. They provide repeatability of the experiment and must be the evidences for detection of stripe rust resistance. 

4)   Line 259: Why lower complementarity is related to higher translocation frequency?

Corrections recommended:

Line 17: … crossed with …

Line 24: … using deletion lines …

Line 34: The areas affected with stripe rust …

Line 35: … many countries … (delete ‘other’)

Line 47: …, and powdery mildew …

Line 56: … possessing …

Line 61: … are examined to determine …  

Lines 67-68: … Oligo-B11, Oligo-Db12H, Oligo-K288, Oligo-pSc119.2, Oligo-pTa535, Oligo-pTa71, Oligo-pSt122, and Oligo-744 were used … (Table S1, Fig. 1). (There are many ‘and’s here.) 

Line 80: … and Oligo-pDb12H showing … (delete ‘for’)

Line 81: The Oligo-744 had …

Line 85: … amphiploid [26], and we found … (add ‘and’)

Line 86: … while the 4Js had …

The explanation to Fig. 1 should be changed and corrected as shown below. Pictures of (a)-(f) do not correspond to the explanations. Bars are not shown in the pictures.

Fig. 1. Sequential ND-FISH patterns of wheat-Th. intermedium double substitution line X479. The probes for FISH are (a) Oligo-k288 (red) + Oligo-D (green), (b) …, (c) …, and (f) …. The Karyotypes of 4Js, 4St and 4St-Js chromosomes are shown using the probe combinations, Oligo-B11 + Oligo-pDb12H (left), Oligo-pSc119.2 + Oligo-pTa535 (middle), and Oligo-744 (right) in (g). 

Line 105: … offspring of the hybrid between X479 and MY11 were …

Line 106: … multiple probes, Oligo- …

Line 109: … chromosome recombinations, of which 26 plants contained telosomic chromosomes, while 51 had chromosome translocations, and one had telosomics and translocations. 

Line 112: … investigated for chromosome rearrangement. (delete ‘analysis the’)

Line 113: shown in Fig. S1, a total of 20% plants contained 63 types of …

Line 114: … 25 different types of broken and translocated chromosomes were ….

Line 115: Among them, translocations occurred higher between … than those between wheat and 1St(25.40%). 

Line 120: … for precise localization of the ….

Line 123: Explanation for Figure 2 is not exactly mentioned. There are six pairs of pictures (a-l) without explanations. In the text, it is written that they are the pictures of Fplants, but in the explanation of Fig. 3, they are F3 plants. Which is correct?

Line 128: … 1BS.1StL and 1StS.1BL were obtained in 11-25% of the plants.

Line 130-132: The M971 had … chromosomes with the transmission …, while M971 had … with 100% transmission …. It indicates that ….

Line 134: … M942 having … M965 having …

Line 136: … using the probe … on the homologous group 1 ….

Line 141: … than those with 4JSL …. 

Line 144: (Showing the translocations is primarily important rather than the probes. Therefore, the explanation should be changed as follows.)

Figure 3. Sequential ND-FISH patterns showing wheat-Th. intermedium homozygous 1St translocation lines, (a, b): 1BS.1StL, and (c, d): 1StS.1BL. The probes for FISH are (a) …, (b) …, (c) …, and (d) ….

Line 150: (Here also, indicating the translocation should be first and then names of the probes.) 

Figure 4. Sequential ND^FISH and Oligo-FISH painting of wheat-4JSL… translocation lines, (a, b) 1DS.4JSL and (c, d) 1BS.4JSL, using the ND-FISH probes (a, c) Oligo-pSc119.2 + Oligo-pTa535 and Oligo-FISH painting probes (b, d) Synt1 + Synt4. The chromosomes indicated with arrows are extracted and put in the boxes to indicate the translocated chromosomes.

Line 155: … wheat centromere specific repeat …

Line 157: … two pairs of …

Line 159: … between the species of wheat and Th. intermedium.

Line 161: … signals …

Line 166: … the metaphase chromosomes of the translocation lines ….

Line 167: … conducted for relative amount to that …

Line 169: … was higher than those of 4BS.4JSL ….

Line 170: … translocations may recruit enough content …

Line 171: … centromere regions …

Line 175: …. (a) Sequential FISH …, and (b) Oligo-pTa71, Oligo-St122 and Oligo CCS1. The arrows indicate the absence of the signals of …. (c) Sequential … and (d) ND-FISH of …. The arrows indicate the presence of Cenh3 signals in … chromosomes. (e) The FISH and …. The numbers 1-3 represent the combinations of ….

Line 183: The Oligo-pDb12H sequence was used for BLAT searching of the ….

Line 184: The JS-chromosomes in the  ….

Line 187: … had 2592 copies. 

Line 189: … were physically localized …

Line 192: The DNAs of the lines …

Line 195: … could be produced in M931, …

Line 200: Figure 6. Physical location of the break point of … deletion detected by FISH and molecular markers. (a) …. Arrow shows …. (b) The PCR … lines (1) CS, (2) MY11, (3) X479 and (4) M931. The stars … amplifications. (Delete ‘The specific … stars.’) (c) The 4VL …. The break point in the 4JSL is shown by an arrow.

Line 212: The symptoms of X479 and MY11 are not shown in Figure 7. 

Line 216: What is ‘IT=0 to 0;)’?

Line 226: … have been showing significant impacts …

Line 232: … revealed the translocations of St-JS chromosomes in X479. 

Line 235: translocations of …

Line 238: … replaced by the Oligo-744 …

Line 242: … in wheat background …

Line 244: … in wheat background …

Line 252: … in different … (delete ‘with’) … in wheat.

Line 256: … double monosomics for …. … in the hybrids …

Line 257: … generated mostly monosomic chromosomes … opportunities of translocations in the ….

Line 259: The types of translocations occurred between wheat chromosomes and 4St-JS were higher than those between wheat chromosomes and 1St, indicating that the complementarity of …. (Why lower complementarity is related to higher translocation?)

Line 261: The broken and fused wheat-Thinopyrum ….

Line 268: … repeats change extremely rapidly ….

Line 269: … containing …

Line 271: … displayed chromosomal …

Line 272: generated distinct signals …

Line 273: … 4BS.4JSL, which were detected by …

Line 274: …. The observed CENH3 signals were co-localized …

Line 275: in wheat with roughly identical densities, …

Line 276: D genome showed …

Line 284: Up to now, more than 84 …

Line 286: …, only Yr50 was transferred from …

Line 289: … translocation line by FISH …

Line 291: and 2JS enhanced the resistance …

Line 295: delete ‘comparison … wheat parent’

Line 301: … with the aim to

Line 306: …, as well as wheat cultivars …

Line 314: digested with 1% …

Line 315: … solution at 37 for 50 min.

Line 317: … with synthesized …

Line 320: … Oligos were predicted …

Line 326: … washed twice with 0.1% Tween 20 in 20xSSC for each 5 min to remove …

Line 327: … painting probed with the …

Line 333: … and their derived …

Line 344: … detection using the goat …

Line 361: … translocations and …

Line 362: … enabled physical localization of … Th. intermedium (Italic) ….

Supplemental Table and Figures

Line 1: … Thinopyrum (Italic) …

Figure S1. The representing chromosome translocations in the F3 progenies of the hybrid between X479 and MY11. The numbers 1-7 indicate the probes, (1) Oligo-pSc119.2, (2) OligopTa535, (3) ….

Figure S2. Distribution of Oligo-pDb12H on the genome of …. (a) Physical distribution of  … on the JS genome, and (b) the copy number comparison of Oligo-pDb12H among the St, J, and JS chromosomes of ….

Figure S3. …. The probes for FISH are (a) …, and (b) …. The cut-pasted chromosomes are 4St-JSS.4JSLdel.

Author Response

The present paper describes cytological identification of the progenies of wheat-Thinopyrum intermedium chromosome double substation line and localization of the gene for stripe rust resistance on the translocated chromosome. The reviewer considers that the quality of the experiment is valuable to be published in Plants. However, many mistakes and wrong usages of English words are found in the manuscript. 

1) The reviewer pointed out the mistakes and gives corrections recommended as below, and would like to ask the authors to correct the manuscript and submit English review to check the expressions.

Response: Thanks for the suggestion. We asked the Dr. Ian Dundas, an expert for wheat molecular cytogenetics and breeding at University of Adelaide, Australia for carefully editing the language in the manuscript.

2) Explanations for Figures should be checked precisely. It is important to indicate the translocated chromosomes. Especially, explanation for Figure 2 is not exactly mentioned. There are six pairs of pictures (a-l). However, there are no explanations on that. In the text, it is written that they are the pictures of Fplants, but in the explanation of Fig. 3, they are F3 Which is correct?

Response: We revised the mistakes according the suggestion. The caption for Figure 2 were corrected for each two pictures for one line. The Fplants are for Fig. 2, and Fplants are for Fig. 3.

3) Line 212: The symptoms of X479 and MY11 are not shown in Figure 7 as mentioned in the text.

Response: We agree to add the symptoms of X479 and MY11 in Figure 7 at the left side.

4) On the markers closely linked to the break point of the translocation, the sequences of the markers used in Fig. 6 for PCR amplification should be shown. They provide repeatability of the experiment and must be the evidences for detection of stripe rust resistance. 

Response: We agree to add the detail primer sequences of the PCR markers in Table S2.

5) Line 259: Why lower complementarity is related to higher translocation frequency?

 Response: The high complementarity chromosomes such as homoeologous group 1, the substitution between wheat and Thinopyrum could be transmitted in the progenies. The broken chromatin form lower complementarity chromosomes may form a more fit translocation to be transmitted in the later generation, since their substitution are not somewhat competitive well.

.

Corrections recommended:

Line 17: … crossed with …

Line 24: … using deletion lines …

Line 34: The areas affected with stripe rust …

Line 35: … many countries … (delete ‘other’)

Line 47: …, and powdery mildew …

Line 56: … possessing …

Line 61: … are examined to determine …  

Lines 67-68: … Oligo-B11, Oligo-Db12H, Oligo-K288, Oligo-pSc119.2, Oligo-pTa535, Oligo-pTa71, Oligo-pSt122, and Oligo-744 were used … (Table S1, Fig. 1). (There are many ‘and’s here.) 

Line 80: … and Oligo-pDb12H showing … (delete ‘for’)

Line 81: The Oligo-744 had …

Line 85: … amphiploid [26], and we found … (add ‘and’)

Line 86: … while the 4Js had …

The explanation to Fig. 1 should be changed and corrected as shown below. Pictures of (a)-(f) do not correspond to the explanations. Bars are not shown in the pictures.

Fig. 1. Sequential ND-FISH patterns of wheat-Th. intermedium double substitution line X479. The probes for FISH are (a) Oligo-k288 (red) + Oligo-D (green), (b) …, (c) …, and (f) …. The Karyotypes of 4Js, 4St and 4St-Js chromosomes are shown using the probe combinations, Oligo-B11 + Oligo-pDb12H (left), Oligo-pSc119.2 + Oligo-pTa535 (middle), and Oligo-744 (right) in (g). 

Line 105: … offspring of the hybrid between X479 and MY11 were …

Line 106: … multiple probes, Oligo- …

Line 109: … chromosome recombinations, of which 26 plants contained telosomic chromosomes, while 51 had chromosome translocations, and one had telosomics and translocations. 

Line 112: … investigated for chromosome rearrangement. (delete ‘analysis the’)

Line 113: shown in Fig. S1, a total of 20% plants contained 63 types of …

Line 114: … 25 different types of broken and translocated chromosomes were ….

Line 115: Among them, translocations occurred higher between … than those between wheat and 1St(25.40%). 

Line 120: … for precise localization of the ….

Line 123: Explanation for Figure 2 is not exactly mentioned. There are six pairs of pictures (a-l) without explanations. In the text, it is written that they are the pictures of Fplants, but in the explanation of Fig. 3, they are F3 plants. Which is correct?

Line 128: … 1BS.1StL and 1StS.1BL were obtained in 11-25% of the plants.

Line 130-132: The M971 had … chromosomes with the transmission …, while M971 had … with 100% transmission …. It indicates that ….

Line 134: … M942 having … M965 having …

Line 136: … using the probe … on the homologous group 1 ….

Line 141: … than those with 4JSL …. 

Line 144: (Showing the translocations is primarily important rather than the probes. Therefore, the explanation should be changed as follows.)

Figure 3. Sequential ND-FISH patterns showing wheat-Th. intermedium homozygous 1St translocation lines, (a, b): 1BS.1StL, and (c, d): 1StS.1BL. The probes for FISH are (a) …, (b) …, (c) …, and (d) ….

Line 150: (Here also, indicating the translocation should be first and then names of the probes.) 

Figure 4. Sequential ND^FISH and Oligo-FISH painting of wheat-4JSL… translocation lines, (a, b) 1DS.4JSL and (c, d) 1BS.4JSL, using the ND-FISH probes (a, c) Oligo-pSc119.2 + Oligo-pTa535 and Oligo-FISH painting probes (b, d) Synt1 + Synt4. The chromosomes indicated with arrows are extracted and put in the boxes to indicate the translocated chromosomes.

Line 155: … wheat centromere specific repeat …

Line 157: … two pairs of …

Line 159: … between the species of wheat and Th. intermedium.

Line 161: … signals …

Line 166: … the metaphase chromosomes of the translocation lines ….

Line 167: … conducted for relative amount to that …

Line 169: … was higher than those of 4BS.4JSL ….

Line 170: … translocations may recruit enough content …

Line 171: … centromere regions …

Line 175: …. (a) Sequential FISH …, and (b) Oligo-pTa71, Oligo-St122 and Oligo CCS1. The arrows indicate the absence of the signals of …. (c) Sequential … and (d) ND-FISH of …. The arrows indicate the presence of Cenh3 signals in … chromosomes. (e) The FISH and …. The numbers 1-3 represent the combinations of ….

Line 183: The Oligo-pDb12H sequence was used for BLAT searching of the ….

Line 184: The JS-chromosomes in the  ….

Line 187: … had 2592 copies. 

Line 189: … were physically localized …

Line 192: The DNAs of the lines …

Line 195: … could be produced in M931, …

Line 200: Figure 6. Physical location of the break point of … deletion detected by FISH and molecular markers. (a) …. Arrow shows …. (b) The PCR … lines (1) CS, (2) MY11, (3) X479 and (4) M931. The stars … amplifications. (Delete ‘The specific … stars.’) (c) The 4VL …. The break point in the 4JSL is shown by an arrow.

Line 212: The symptoms of X479 and MY11 are not shown in Figure 7. 

Line 216: What is ‘IT=0 to 0;)’?

Line 226: … have been showing significant impacts …

Line 232: … revealed the translocations of St-JS chromosomes in X479. 

Line 235: translocations of …

Line 238: … replaced by the Oligo-744 …

Line 242: … in wheat background …

Line 244: … in wheat background …

Line 252: … in different … (delete ‘with’) … in wheat.

Line 256: … double monosomics for …. … in the hybrids …

Line 257: … generated mostly monosomic chromosomes … opportunities of translocations in the ….

Line 259: The types of translocations occurred between wheat chromosomes and 4St-JS were higher than those between wheat chromosomes and 1St, indicating that the complementarity of …. (Why lower complementarity is related to higher translocation?)

Line 261: The broken and fused wheat-Thinopyrum ….

Line 268: … repeats change extremely rapidly ….

Line 269: … containing …

Line 271: … displayed chromosomal …

Line 272: generated distinct signals …

Line 273: … 4BS.4JSL, which were detected by …

Line 274: …. The observed CENH3 signals were co-localized …

Line 275: in wheat with roughly identical densities, …

Line 276: D genome showed …

Line 284: Up to now, more than 84 …

Line 286: …, only Yr50 was transferred from …

Line 289: … translocation line by FISH …

Line 291: and 2JS enhanced the resistance …

Line 295: delete ‘comparison … wheat parent’

Line 301: … with the aim to

Line 306: …, as well as wheat cultivars …

Line 314: digested with 1% …

Line 315: … solution at 37℃ for 50 min.

Line 317: … with synthesized …

Line 320: … Oligos were predicted …

Line 326: … washed twice with 0.1% Tween 20 in 20xSSC for each 5 min to remove …

Line 327: … painting probed with the …

Line 333: … and their derived …

Line 344: … detection using the goat …

Line 361: … translocations and …

Line 362: … enabled physical localization of … Th. intermedium (Italic) ….

Supplemental Table and Figures

Line 1: … Thinopyrum (Italic) …

Figure S1. The representing chromosome translocations in the F3 progenies of the hybrid between X479 and MY11. The numbers 1-7 indicate the probes, (1) Oligo-pSc119.2, (2) OligopTa535, (3) ….

Figure S2. Distribution of Oligo-pDb12H on the genome of …. (a) Physical distribution of  … on the JS genome, and (b) the copy number comparison of Oligo-pDb12H among the St, J, and JS chromosomes of ….

Figure S3. …. The probes for FISH are (a) …, and (b) …. The cut-pasted chromosomes are 4St-JSS.4JSLdel.

Response: Thank you so much for the careful correction. We agree to accept all the points and revised accordingly.  The gramatic errors were corrected in overall text of the manuscript and supplementary materials.

Reviewer 3 Report

This manuscript presents an interesting study on the development of wheat-Thinopyrum substitution lines for stripe rust resistance.

In general, the manuscript is well written and it can be to interest for the readers of this journal.

Nevertheless, some minor changes should be carried out.

L14: The Latin name of the species (Th. intermedium) should be in italics.

L31: The more accepted Latin name for common wheat is ‘Triticum aestivum L. ssp. aestivum’.

L43: The first time that a species is named, the complete Latin notation, including author, should be indicated.

The line X479 or any other should be always named equal. Thought the text, the authors change the notation of this material between ‘line X479’ and ‘X479’.

L211-212: The authors indicate that the resistance response was measured in line X479 and its derivates. However, the Figure 7 shows some lines but not the line X479 as the authors suggest in the text. This should be clarified.

L305: The sentence should begin with capital letter. Furthermore, the reference 18 is duplicated and should only be written according with the journal style.

L357: ‘line X479’ instead of ‘lin X479’.

L363: Newly, the Latin name of the specie should be written in italic.

Author Response

This manuscript presents an interesting study on the development of wheat-Thinopyrum substitution lines for stripe rust resistance.

In general, the manuscript is well written and it can be to interest for the readers of this journal.

Nevertheless, some minor changes should be carried out.

L14: The Latin name of the species (Th. intermedium) should be in italics.

Response: We agree to the correction.

L31: The more accepted Latin name for common wheat is ‘Triticum aestivum L. ssp. aestivum’.

Response: We agree to the corrections.

L43: The first time that a species is named, the complete Latin notation, including author, should be indicated.

Response: We agree to the corrections.

The line X479 or any other should be always named equal. Thought the text, the authors change the notation of this material between ‘line X479’ and ‘X479’.

Response: We agree to the corrections and revised accordingly overall the manuscript.

L211-212: The authors indicate that the resistance response was measured in line X479 and its derivates. However, the Figure 7 shows some lines but not the line X479 as the authors suggest in the text. This should be clarified.

Response: We agree to add the leaves' symptoms of X479 and MY11 in Figure 7 at the left side.

L305: The sentence should begin with capital letter. Furthermore, the reference 18 is duplicated and should only be written according with the journal style.

Response: We agree to the correction.

L357: ‘line X479’ instead of ‘lin X479’.

Response: We agree to the correction.

L363: Newly, the Latin name of the specie should be written in italic.

 Response: We agree to the correction and checked all the names of the entire manuscript.